# Multi-dimensional super-resolution imaging enables surface hydrophobicity mapping

Marie N. Bongiovanni[1,*], Julien Godet[2,*], Mathew H. Horrocks[1,*], Laura Tosatto[1,3], Alexander R. Carr[1], David C. Wirthensohn[1], Rohan T. Ranasinghe[1], Ji-Eun Lee[1], Aleks Ponjavic[1], Joelle V. Fritz[4], Christopher M. Dobson[1], David Klenerman[1] & Steven F. Lee[1]

Super-resolution microscopy allows biological systems to be studied at the nanoscale, but has been restricted to providing only positional information. Here, we show that it is possible to perform multi-dimensional super-resolution imaging to determine both the position and the environmental properties of single-molecule fluorescent emitters. The method presented here exploits the solvatochromic and fluorogenic properties of nile red to extract both the emission spectrum and the position of each dye molecule simultaneously enabling mapping of the hydrophobicity of biological structures. We validated this by studying synthetic lipid vesicles of known composition. We then applied both to super-resolve the hydrophobicity of amyloid aggregates implicated in neurodegenerative diseases, and the hydrophobic changes in mammalian cell membranes. Our technique is easily implemented by inserting a transmission diffraction grating into the optical path of a localization-based super-resolution microscope, enabling all the information to be extracted simultaneously from a single image plane.

[1] Department of Chemistry, University of Cambridge, Lensfield Road, Cambridge CB2 1EW, UK. [2] Laboratoire de Biophotonique et Pharmacologie, UMR CNRS 7213, Université de Strasbourg, Faculté de Pharmacie, 67400 Illkirch, France. [3] DTI Laboratory of Neurodegenerative Diseases, Centre for Integrative Biology Università degli Studi di Trento, via Sommarive 9, Trento 38123, Italy. [4] Luxembourg Centre for Systems Biomedicine, University of Luxembourg, Campus Belval, 7 avenue des Hauts-Fourneaux, Esch-sur-Alzette, Luxembourg L-4362, Luxembourg. * These authors contributed equally to this work. Correspondence and requests for materials should be addressed to S.F.L. (email: sl591@cam.ac.uk).

Super resolution (SR) microscopy is an advanced optical imaging technique that retains the advantages of fluorescence imaging, whilst enabling the resolution of structures at length scales that are particularly relevant to biological activities. Despite their current successes, SR techniques[1–3] have been focused on reading out positional information from the intensity of fluorescence emission; however, other optical properties, such as the fluorescence emission spectrum, remain under-utilized.

Spectrally-resolved single-molecule fluorescence experiments in the condensed phase have previously found widespread application, including the direct observation of photo-oxidation in nanocrystals[4], multiple colour or spectral-demixing labeling[5], pattern-matching techniques for efficient identification of fluorophore ratios[6] and dissecting the architecture of multiprotein complexes[7,8], although these studies have generally been confined to point-source measurements. These single-molecule microscopy methods have taken advantage of the multi-dimensionality of the fluorescence properties, such as anisotropy, fluorescence lifetime, fluorescence quantum yield and inter-fluorophore distance[4,8,9]. However, these properties have remained largely unexplored for SR imaging applications. While simultaneous spectral and spatial imaging for localization microscopy has been described[10–12], these methods do not take into account the environmental-specific properties of the dye. Nevertheless, it is possible for individual dye molecules to be compatible with SR imaging and also to encode those fluorescence properties in their nano-environment. This general principle, of simultaneously extracting positional and environmental-specific hyper-spatial information (which we define as any orthogonal fluorescence not in the spatial or temporal domain) we herein describe as multi-dimensional super-resolution (md-SR) imaging. All that is required for md-SR imaging is a fluorophore that varies its emission wavelength depending on its environment, and that can be made to undergo fluorescence intermittency to isolate single emitters.

Here, we report the development of an md-SR imaging technique termed spectrally-resolved PAINT (points accumulation for imaging in nanoscale topography[13]) or sPAINT, which simultaneously records the spatial position and emission spectrum of single dye molecules to super-resolve an image. sPAINT can generate information-enriched md-SR images through the use of spectrally-responsive fluorophores. For example, the fluorescence emission wavelength of the phenoxazone-based dye nile red (NR) is known to be sensitive to the hydrophobicity of its environment[14–18], and we have exploited this with sPAINT to super-resolve biological structures in the hydrophobicity domain.

## Results

**sPAINT enables single-molecule spectroscopy**. In sPAINT, the spatial and spectral information of the fluorescent molecules were simultaneously collected to generate multi-dimensional datasets $(x,y,\lambda)$. sPAINT was easily implemented using a single-molecule fluorescence imaging microscope with the addition of a blazed transmission diffraction grating placed before the image plane (Fig. 1a; Equations 1 and 2). The two major diffraction orders (zeroth order and first order, Fig. 1a) were projected onto two non-overlapping areas of the same detector and recorded simultaneously. The zeroth order was used to super-localize spatially ($x,y$ coordinates; $m_0$) the diffraction-limited puncta (Fig. 1b, left), while the first order diffraction was used to collect the spectral information ($\lambda$; $m_1$) of the fluorescence signal (Fig. 1b, right).

To validate the setup, multi-dye labeled, diffraction-limited TetraSpeck microspheres (100 nm), with well-defined fluorescence spectra, were used to empirically determine the linearity in the dispersion from the spatial-to-spectral domains (Supplementary Fig. 1), and to correct for aberrations of the grating over the field of view (For more details, see Methods, Supplementary Fig. 3, and Equation 3). The emission spectra of the individual beads projected onto the EMCCD chip correlated well with the bulk fluorescence spectrum of the TetraSpeck beads and was linear over a wide wavelength range (Fig. 1c, Pearson correlation of 0.92 between wavelengths of 550–750 nm). As these diffraction-limited beads act as point emitters, there was no need for a mechanical slit, which is otherwise used to control the spectral resolution in conventional spectrometers.

The need to super-resolve structures means that sPAINT images are typically limited by the spatial stability of the diffraction-limited point source, not the spectral resolution. We selected a transmission grating which asymmetrically divides the fluorescence emission by diffracting ∼60% into the zeroth order (spatial domain), with the remaining ∼40% being used to resolve the whole emission spectrum. We then empirically determined the spatial stability by measuring the lateral displacement of super-localized fluorescence puncta from single diffraction-limited beads as a function of detected photon number using a previously described methodology[19] and determined an ultimate spatial precision of the instrument (that is with an infinite photon number) of $6.6 \pm 0.2$ nm, typically peak-values obtained during sPAINT (∼800 photons) led to a spatial-localization precision of ∼18 nm (Fig. 1d, Supplementary Fig. 4, see Methods). In addition, for each localization event, the displacement in the center of the emission spectrum was measured simultaneously, thereby relating the photons detected in the spatial domain to the spectral precision of the instrument. The ultimate spectral precision was empirically determined to be $1.3 \pm 0.1$ nm (Fig. 1e) with typical peak values obtained during sPAINT (∼800 photons in the spatial domain) led to a spectral localization precision of ∼3.8 nm.

The dye NR (Fig. 1f) was selected for sPAINT imaging due to its lipophilic nature, its shift in absorption during transient binding, and more importantly, its wavelength of fluorescence emission is highly dependent on the local hydrophobicity of its environment[14,16,20]. Therefore NR is an excellent candidate for sPAINT imaging, since it can act as both a PAINT dye and a spectrally–responsive probe of the local hydrophobic environment. Due to the low signal in the spectral domain (as discussed above), the center of a Gaussian function was used to approximate the center position of the peak emission spectrum of individual NR molecules in sPAINT experiments (Supplementary Fig. 2 and Methods – for extended discussion of fit parameters and quality control).

**Nile red resolves LUV hydrophobicity**. Large unilamellar vesicles (LUVs) are well-characterized synthetic membrane mimics, whose lipophilicity can be readily altered by varying the lipid composition[17]. LUVs (∼100 nm, Supplementary Table 1) were therefore used to characterize the spectrally-resolved single-molecule properties of NR for sPAINT. Thousands of NR single-molecule spectra from hundreds of spatially isolated LUVs (Supplementary Fig. 5), composed of three distinct lipid compositions (Fig. 2a), were measured including those formed from: 1,2-Dioleoyl-sn-glycero-3-phosphocholine (DOPC), sphingo-myelin (N-(tricosanoyl)-sphing-4-enine-1-phosphocholine, SM) and a mixture of sphingomyelin and cholesterol (SM + CL, 2:1 molar ratio). These three lipid compositions were chosen to represent different phase states (Supplementary Table 1)[17]. Single

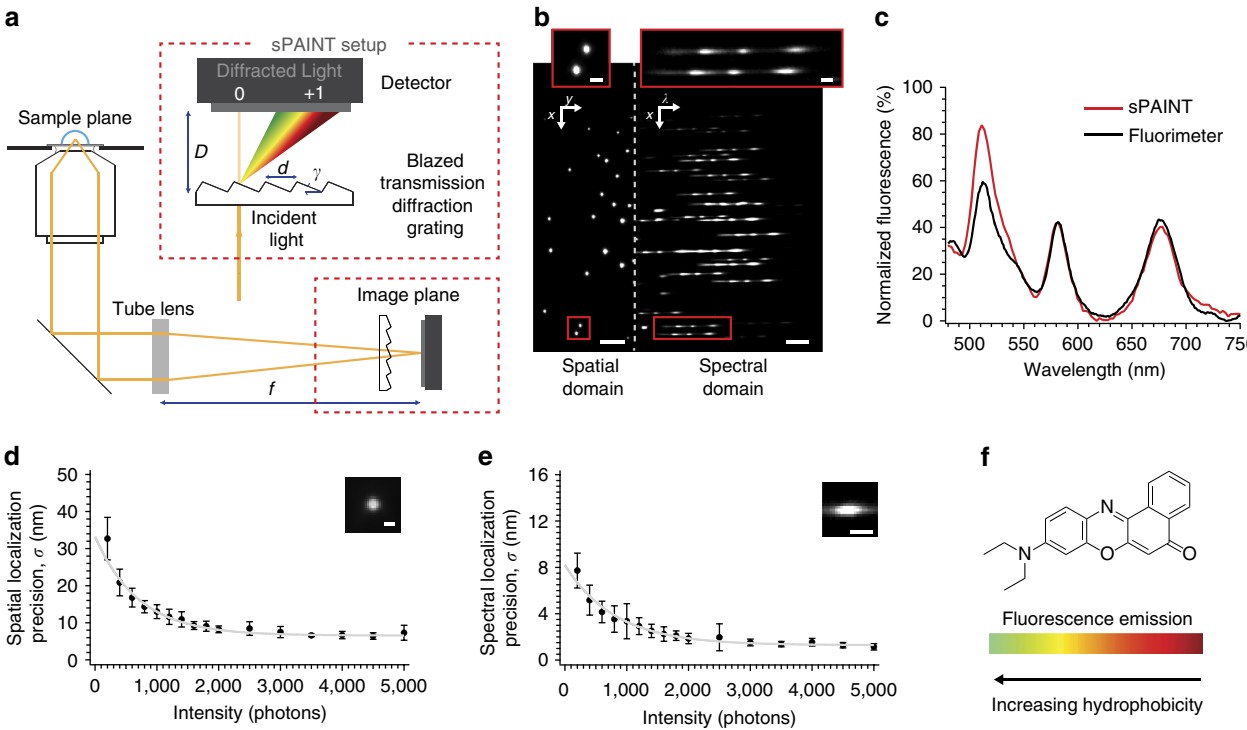

**Figure 1 | sPAINT instrumental setup.** (**a**) Scheme of the sPAINT setup; single-molecule fluorescence is collected by a high NA objective lens and focused by the tube lens before passing through a blazed transmission diffraction grating. Here the fluorescence emission is divided into either the spatial region (*x,y* space, zeroth order diffraction) or the spectral region (λ space, first order diffraction) in the image plane and recorded on a single chip of an EMCCD camera; where, *D* is the grating-to-detector distance (~20 mm), *d* is the grating line spacing (300 groves per mm), γ is blaze angle (8.6°) and *f* is the focal length of the tube lens (~200 mm). (**b**) Contrast adjusted, representative raw sPAINT data of 100 nm TetraSpeck microspheres – labelled with four different fluorophores – demonstrating both the zeroth order diffraction spatial region (left; *x,y*) and first order diffraction spectral region (right; λ). Scale bars are 5 µm (spatial) and 100 nm (spectral). A magnified inset of TetraSpeck beads with the corresponding emission spectrum is included as an inset (red). Scale bars are 1 µm (spatial) and 20 nm (spectral). (**c**) Comparison of fluorescence intensity versus wavelength for a single sPAINT bead and bulk fluorimeter data. (**d**) Empirically determined spatial precision, ($\sigma_{xy}$) (*n* = 1,000, number of beads used for calibration). (**e**) Empirically determined spectral precision ($\sigma_{\lambda}$) (mean ± s.d., *n* = 400 beads). (Supplementary Fig. 4). (**d,e**) The photon values refer to the same spatial location. (**f**) The chemical structure of nile red and corresponding fluorescence emission wavelength dynamic range in typical hydrophobic environments.

NR molecules displayed similar localization precision when compared with the data obtained from multi-dye labelled beads for the same integrated photon number (Supplementary Fig. 6 c.f. Fig. 1d,e) with the ultimate spatial precision of NR being slightly lower than beads (~6 versus 8 nm, respectively). This enabled the LUVs to be resolved at diffraction-limited (DL), intensity-based SR and sPAINT levels (Fig. 2b). As expected, we were not only able to super-resolve the physical size of the LUVs (Supplementary Fig. 7), but also the different hydrophobic environments of each LUV was easily differentiated as shown in the false-colour sPAINT images, observed as red, yellow and blue for DOPC, SM and SM + CL LUVs, respectively (Fig. 2b, for additional examples see Supplementary Fig. 5). The full distribution of the NR emission spectrum was determined by collating individual NR spectral positions (Fig. 2c), and the peak spectral position was indistinguishable from bulk fluorimeter values (Supplementary Fig. 8). In addition, a linear relationship in the mean locations per LUV per unit time was observed for all three lipid compositions, indicating that NR undergoes conventional PAINT in all cases, albeit preferentially for some environments more than others (Fig. 2d).

As the least hydrophobic environment was DOPC, we used LUVs composed of DOPC to determine the maximum typical concentration of NR for applications in more complex sPAINT experiments. The number of localizations per unit area of lipid (localization density) was plotted as a function NR concentration, giving rise to a steady increase until a maximum was obtained at

~50 nM (Fig. 2e). By determining the residency time and localization rate of each NR molecule (Fig. 2f; Supplementary Fig. 9), it was found that at concentrations of NR > 50 nM, a large increase in the mean residency time was observed, which was attributed to multiple NR binding events in the same time-bin; these data were supported by multi-state intensity trajectories (Fig. 2g). Therefore, effective sPAINT experiments should be performed at NR concentration ~50 nM, although this will vary depending upon the exact lipid composition/sample type.

**sPAINT quantifies hydrophobicity of single protein aggregates.** Protein misfolding disorders, such as Parkinson's and Alzheimer's disease are associated with the deposition of aggregates of the proteins α-synuclein (αS) and amyloid-β (aβ), respectively. The aggregation of the protein results in populations of mature amyloid fibrils and immature aggregates, such as oligomers. Interestingly, the surface hydrophobicity, or solubility of the protein aggregates associated with these diseases is thought to be central to their toxicity[21,22]. sPAINT was therefore used to determine the hydrophobicity of individual protein aggregates assembled *in vitro*. Hundreds of spatially isolated fibrils and oligomers composed of either αS, or aβ$_{1-42}$ were resolved at the DL, SR and sPAINT levels (Representative images shown in Fig. 3a–d, Supplementary Fig. 10). As expected, mature aggregates from both proteins appeared with a fibrillar morphology (Fig. 3b,d, Supplementary Fig. 11), and oligomers appeared as super-resolved amorphous puncta (Fig. 3a,c).

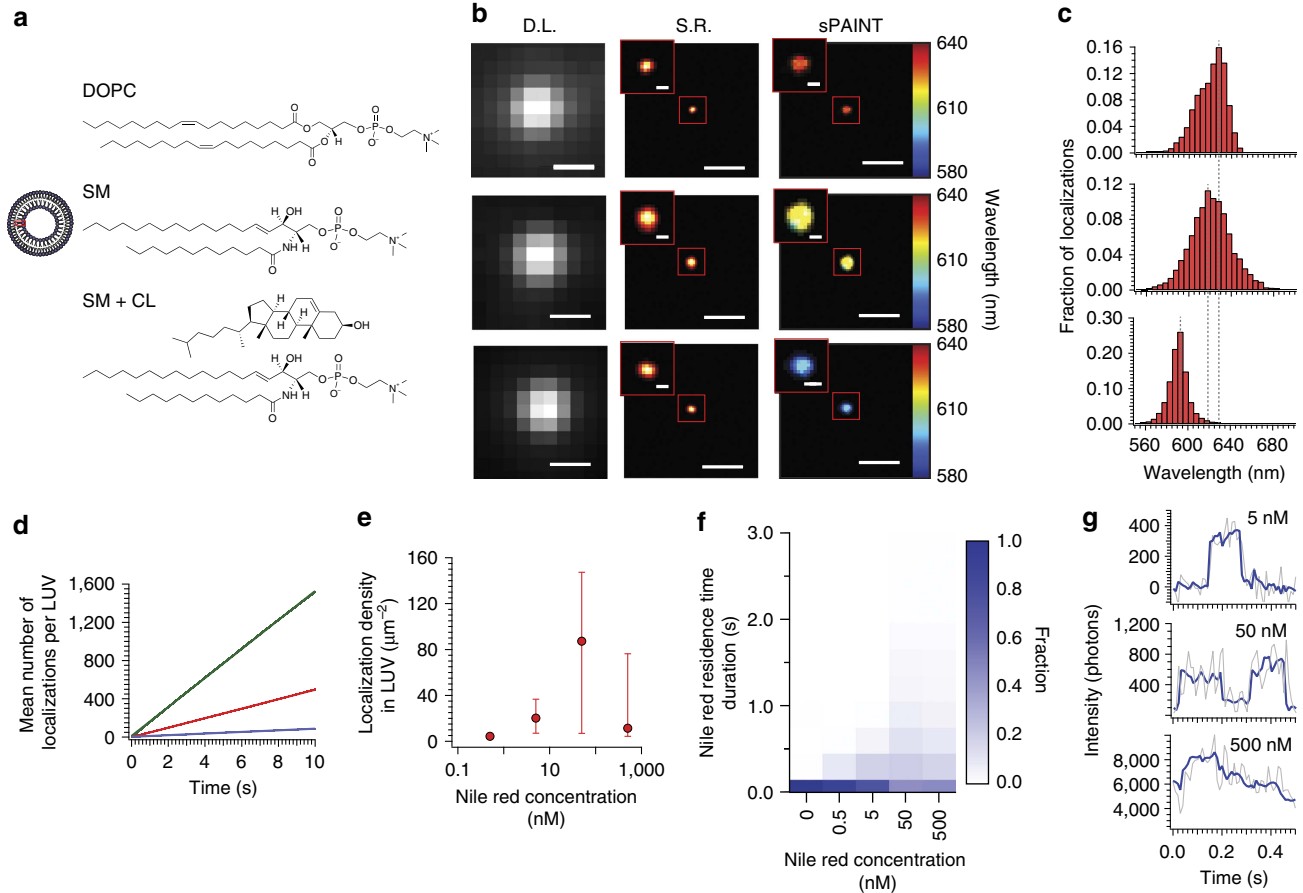

**Figure 2 | Nile Red sPAINT calibrated on synthetic 100 nm unilamellar vesicles (LUVs). (a)** Scheme of the LUV and rows from top-to-bottom; LUVs composed of DOPC lipid, SM lipid or SM/cholesterol lipid. **(b)** Columns from left-to-right; diffraction-limited image (D.L.), super-resolution image (S.R.), sPAINT hydrophobicity map and **(c)** frequency histogram of fluorescence emission peak (peak photon values above background were typically ~750; DOPC, $n = 42,559$ ($N = 542$); SM, $n = 5,968$ ($N = 154$); SM + CL, $n = 48,589$ ($N = 102$); where $n$ is the number of spectrum centres in the histogram and $N$ is the number of LUVs. **(d)** NR localization rate as a function of lipid type (green line: DOPC, blue line: SM, red line SM + CL). **(e)** NR localization density as a function of NR concentration. Error bars show the inter-quartile range for at least 90 LUVs in each case. **(f)** Mean on-time per NR localization, **(g)** concentrations above 50 nM led to multiple NR binding events within a typical exposure time of the detector (50 ms). Scales bars are 500 nm and 20 nm in zoom.

The ability to image oligomers or fibrils, on an individual basis[23], allows the distribution of sPAINT localizations for each aggregate to be considered and compared with bulk values, for example, the sPAINT distribution from a single, large αS fibril (Fig. 3b – red box) is red-shifted, meaning this fibril is less hydrophobic compared with the fibrils in the sample as a whole.

sPAINT images also revealed a broad distribution of hydrophobicity within individual oligomers of the same protein (Fig. 3a,c), which appears as a mixture of colours (green and yellow) in the sPAINT render. The ability to observe oligomers/fibrils with unique sPAINT signatures is relevant since, at least for αS, multiple structural states have been observed[24,25]. Therefore, sPAINT is a relevant exploratory tool to probe sub-populations and to potentially correlate aggregate structure with hydrophobicity.

When the full distribution of the NR emission spectrum was considered, mature αS fibrils were less hydrophobic than the αS oligomers (Fig. 3f), interestingly the opposite trend was observed for aβ$_{1-42}$ fibrils and oligomers (Fig. 3g). The distribution of aβ$_{1-42}$ oligomers appeared bimodal and much broader than αS oligomers, indicating a wide variation in oligomer types for the aβ$_{1-42}$ protein. The decreased width of the NR distribution observed for αS fibrils (fibrils s.d. is 14 nm, c.f. oligomers s.d. 32 nm, Fig. 3f) was indicative of NR interacting with a uniform

environment, which is consistent with the highly ordered structure of mature fibrils[26]. Direct measurement of the heterogeneous distribution of all these new quantities may reveal details of the underlying toxicity and relevance to disease.

**sPAINT probes hydrophobicity of mammalian cell membranes.** The plasma membrane of mammalian cells is a highly dynamic structure, which is characterized by regions of differing fluidity that arise from lipid and protein organization that play a critical role in cell processes, such as signalling[27]. While techniques are available to monitor the diffusion of molecules to study fluidity[28,29], researchers have yet to address the visualization of hydrophobicity changes[30]. Therefore, sPAINT was next applied to the image the complex environment of the mammalian cell membrane (Fig. 4).

sPAINT was used to spatially and temporally map the hydrophobicity of intact, adherent neuronal-like SH-SY5Y cell plasma membranes. The cell plasma membrane was resolved at the DL, SR and sPAINT levels (Fig. 4a) for both live and fixed cells (Supplementary Fig. 12). When the full distribution of the NR emission spectrum was considered (mean ± s.d. of the wavelength frequency histogram is 602 ± 5.5 nm), it was consistent with NR data from the hepatic microsomal cell plasma

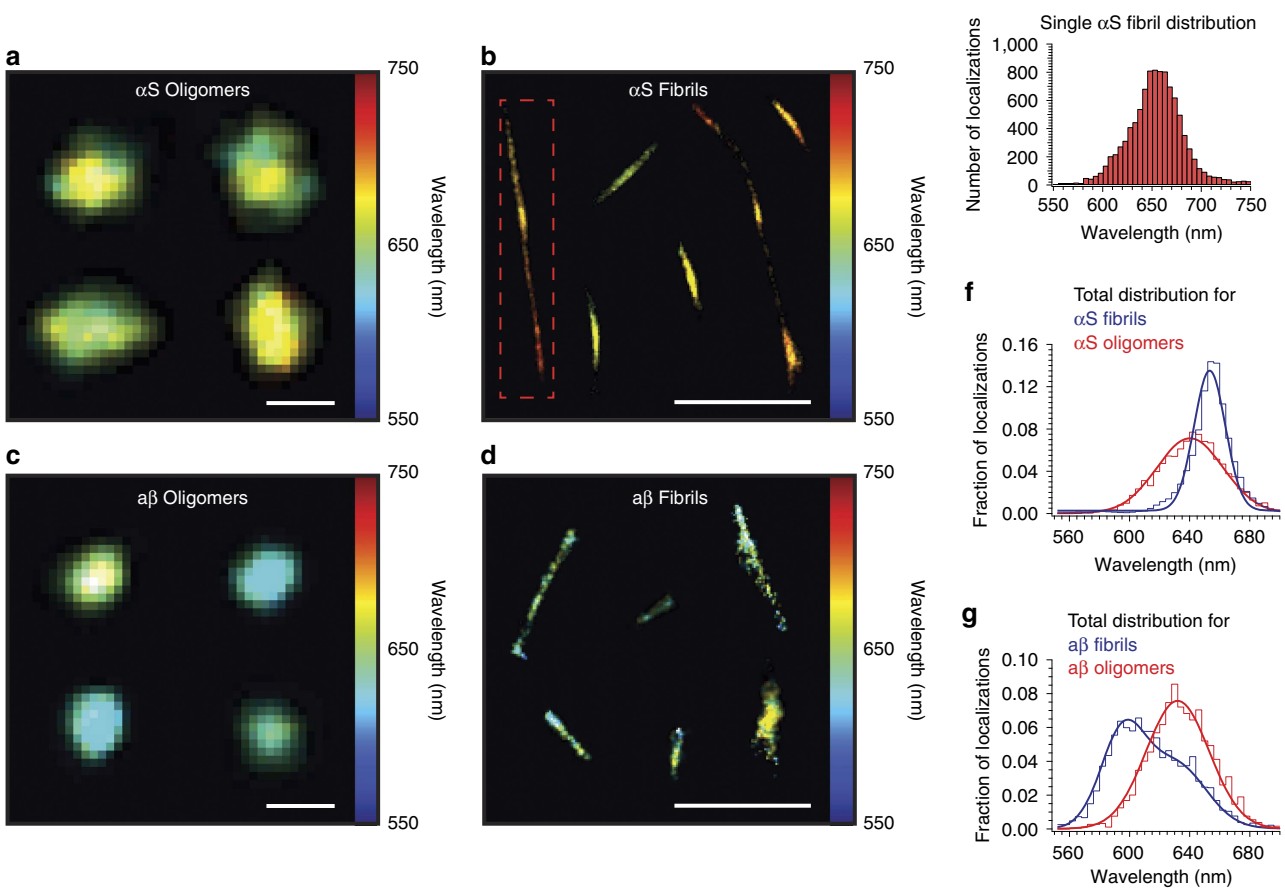

**Figure 3 | sPAINT imaging of protein aggregates associated to disease.** (a) Representative sPAINT hydrophobicity image of single αS oligomers. Scale bar is 100 nm. (b) Representative sPAINT hydrophobicity image of individual αS fibrils. By examining aggregates one-at-a-time, it is possible to extract information about the variation in hydrophobicity of a single fibril (b – red box). Scale bar is 1 μm. (c) Representative sPAINT hydrophobicity image of amyloid-β oligomers. Scale bar is 100 nm. (d) Representative sPAINT hydrophobicity image of individual amyloid-β fibrils. Scale bar is 1 μm. (e) Frequency histogram of the individual sPAINT localizations that are used to generate the fibril (b – red box, n = 9,996 localizations). (f) Total frequency histogram of the individual sPAINT localizations from both αS oligomers (red, n = 17,619 localizations, N = 539 oligomers; peak photon values above background ∼780) and αS fibrils (blue, n = 120,275 localizations, N = 1,528 fibrils; peak photon values above background ∼1,100) – note the shift in bulk hydrophobicity and narrowing of the distributions that are used to generate the fibril (b – red box). (g) Frequency histogram of the individual sPAINT localizations from both aβ$_{1-42}$ oligomers (red, n = 6,133 localizations, N = 80 oligomers; peak photon values above background ∼860) and aβ fibrils (blue n = 17288 localizations, N = 668 fibrils; peak photon values above background ∼710).

membrane that contain a similar plasma membrane lipid composition[14]. In addition, sPAINT was sensitive to alterations in membrane cholesterol levels (Fig. 4b,c) altered using well-established modulating protocols[31]. A red-shift in the mean position and a reduction of the distribution-width of hydrophobic localizations was observed for the membranes of cholesterol-depleted cells compared with untreated cells (Fig. 4c, green line c.f. red line), whereas a blue-shift in the mean position of hydrophobic localizations was observed for cholesterol-enriched membranes (Fig. 4c, blue line c.f. red line). Next, changes in the hydrophobicity were observed temporally, leading to the direct visualization of time-evolved super-resolved 'hotspots' on the plasma membrane (Fig. 4d and Supplementary Movie 1) and displayed a greater than a three fold wider distribution than a non spectrally-sensitive fluorophore[32] (Supplementary Fig. 13). The change in hydrophobicity along the plasma membrane was visualized by applying a Nadaraya–Watson kernel[33] regression, and then a time-moving box of 200 over 3,000 frames to the spectral information (n = 1,175 localizations), where the temporal resolution was defined by a moving box shift of 0.8 s (16 frames). The ability to investigate the hydrophobicity in precise regions of

the cell plasma membrane at the nanoscale (for fixed cells, where the membrane is expected to be minimally fluid[34]) represents the advancement from spectral ratiometric imaging typically applied in the field or in bulk fluorescence measurements[14,35].

## Discussion

md-SR techniques, such as sPAINT, advance SR methods by providing both positional and environmental information to super-resolve biological structures. sPAINT is simple-to-implement, only requiring a single transmission diffraction grating and no additional detectors, or sophisticated optics. sPAINT was applied to map the hydrophobicity of dynamic systems using NR, including cell membranes and protein aggregates, demonstrating the scope of applications.

md-SR can exploit any dye that exhibits both an environmentally-specific spectral shift, and can be individually localized at the single-molecule level. Although, in this study, we have concentrated on the application for sensing hydro-phobicity at the nanoscale, md-SR can be applied to a multitude of other dimensional quantities. For instance, we have already

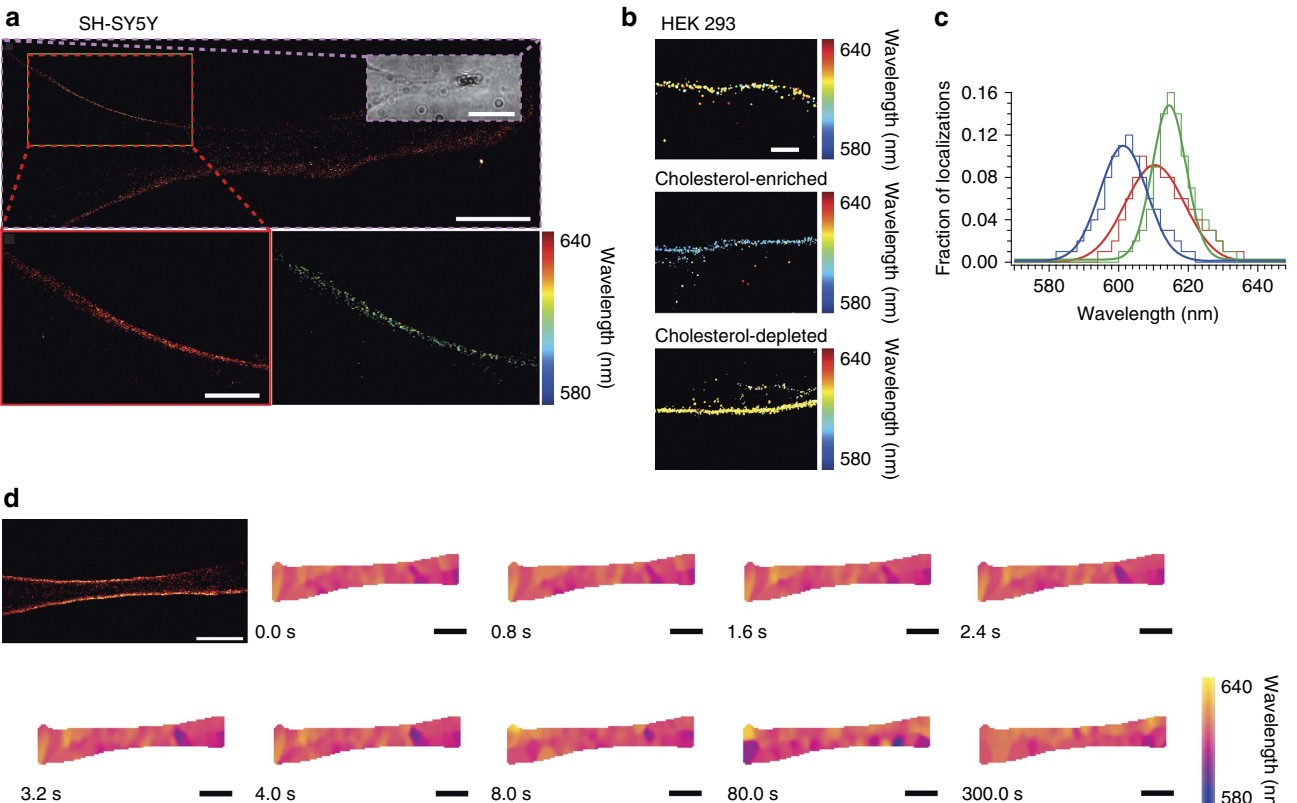

**Figure 4 | sPAINT imaging along the length of the cell plasma membrane.** (**a**) Representative SR image of a live neuron-like SH-SY5Y cell (inset: white-light image). Scale bar is 5 µm (n = 31,744 localizations peak photon values above background ∼543) and corresponding zoom SR (bar is 0.5 µm, n = 2,859 localizations) and sPAINT image (exposure time 50 ms, 3,000 frames). (**b**) HEK 293 cells untreated or cholesterol manipulated. Scale bar is 100 nm. (**c**) Frequency histogram of fluorescence emission peak for cholesterol experiments, red line is untreated cells (n = 18,227 localizations; peak photon values above background ∼700), blue line is cholesterol-enriched cells (n = 27,183 localizations; peak photon values above background ∼990) and green line is cholesterol-depleted cells (n = 20,838 localizations; peak photon values above background ∼640). Data in (**c**) is from ∼10 cells per treatment. (**d**) Spatial-temporal information of fixed epithelial SH-SY5Y cell. Representative SR image and spatial-temporal changes in nile red emission wavelengths (Scale bars are 1 µm, n = 42,099 localizations). Localization metadata generated by sPAINT include localization (x and y coordinates), time (frame number) and spectral (λ) information (Supplementary Video 1). Hydrophobicity maps were generated by applying Nadaraya–Watson kernel regression to the spectral information retrieved at every point localization and observed for a given time interval, with a temporal resolution defined by box-car averaging to ∼0.8 s.

successfully used the dye JC-1, a carbocyanine extensively used to study mitochondrial membrane potential[36–38], to generate md-SR images of protein aggregates[39] (Supplementary Fig 14 a–d). In addition we have used dioxaborine molecular-rotor dyes[40] to generate viscosity-sensitive super-resolved images of diffraction-limited outer membrane vesicles from *Salmonella Typhi* (Supplementary Fig 14 e–h). It should also be possible to monitor multiple properties in the same sample either by using probes that have no spectral overlap, pre imaging/bleaching of non-PAINT probes[41], or by performing sequential imaging rounds interspersed with washing steps to remove each PAINT label[42,43].

There is some inherent loss of photons required for gaining spectral information both in terms of the inefficiency of the transmission grating (∼30%) and a trade off in diverting light from the spatial into the spectral domain. We balanced these spatial and spectral requirements by selecting a diffraction grating that diverted ∼60% of the light to the zeroth order, and ∼40 % to the first order (Fig. 1d,e), since, counter-intuitively, the precision in the spatial domain limits the overall resolution of the technique. In addition, not all of the localizations give rise to both a useable spatial and spectral fit (only ∼40% of the localizations can be used to gain spectral information). Depending on the sample type, this can be offset by the near-infinite temporal

acquisition afforded by the PAINT technique. Furthermore, the sample geometry and orientation should be taken into account for optimal imaging conditions (Supplementary Fig. 15).

In summary, the sPAINT approach is simple to implement and opens up the field of md-SR imaging, having potentially wide use for studying the hydrophobicity of any membrane-bound biological sample, or individual protein aggregates. The applicability of the sPAINT method used to extract the emission spectrum from individual dye molecules is not limited to PAINT experiments, and could be extended for use with other SR techniques, such as dSTORM and PALM. Hydrophobic reporters at the nanoscale are useful tools to better understand processes that drive many biological interactions including: the dynamics at the cellular plasma membrane, or to understand the toxicity of protein aggregates to cells.

## Methods

**Preparation of nile red.** Phosphate-buffered saline (PBS, 00-3002, ThermoFisher scientific) was freshly prepared using high-purity Milli-Q water of resistivity 18 MΩ cm and filtered (0.02 µm cut-off, Whatman) and used without further pH adjustment. Nile red (NR) stock solutions were prepared by dissolving NR into dimethyl sulfoxide (DMSO, D2650 >99.7% purity) and then diluted into pre-filtered PBS buffer and used at a final concentration of 0.05 µM unless otherwise specified.

**Preparation of coverslips for imaging.** Glass coverslips (0.13 mm thickness, 22 × 22 mm, VWR collection, 631-0124) were cleaned using an argon plasma cleaner (PDC-002, Harrick Plasma) for 1 h, then Frame-seal slide chambers (9 × 9 mm, Bio-rad, Hercules, USA) were affixed to the glass coverslips. The chamber was filled with poly-L-lysine solution (0.01% w/v, P4707) to fully coat the cover-glass surface, incubated for at least 30 min and then washed three times with filtered PBS buffer. The slide was transferred to the microscope stage and optically coupled to the objective lens through index-matching immersion oil ($n = 1.518$, Olympus, UK).

**Preparation of lipid vesicles.** Dioleoylphosphatidylcholine (DOPC, Avanti Polar Lipids), dipalmitoylphosphatidylcholine (DPPC, Avanti Polar Lipids), Milk sphingomyelin (SM, Avanti Polar Lipids), Dioleoylphosphatidylserine (DOPS), or cholesterol were used to form large unilamellar vesicles (LUVs) by the extrusion method as previously described[44]. Briefly, a suspension of unilamellar vesicles in PBS (pH 7.4) was extruded (31 passages) by using an Avanti extruder (Avanti Polar Lipids) and a membrane with 0.1 μm pore size (Avanti Polar lipids). This generated monodisperse LUVs with a mean diameter centred at 0.11 ± 0.01 μm as measured with a Malvern Zetamaster 300 (Malvern, UK). LUVs were extruded and imaged immediately. Data are representative of three experiments conducted on separate days.

**Protein expression and purification.** For the αS protein, BL21(DE3) Gold cells (Stratagene) were transformed with wild-type (WT) human αS. Starters were diluted into Overnight Express Instant TB Medium (Novagen) supplemented with 1% glycerol and left to grow for 16–18 h at 30 °C. Cells were then harvested and the protein was purified as previously published[45].

Human amyloid-β 1–42 protein (aβ$_{1-42}$) was purchased from Anaspec (USA). The protein was re-suspended in 10 mM NaOH in MQ water and purified by high-performance-liquid-chromatography (Agilent) using a BIoSep_SEC-s2000 column (OOH-214-K0) 300 × 7.80 mm. The protein was eluted using PBS and peak fractions were collected, frozen immediately in liquid nitrogen and stored at −80 °C until use.

**Preparation of protein aggregates.** WT αS was prepared from gel-filtration purified monomer fractions flash frozen in liquid nitrogen. Monomer was ultra-centrifuge before aggregate formation for 1 h, 90000 r.p.m. at 4 °C using a TLA100 rota. Experiments were conducted in triplicate. For each experiment, 600 μL of 70 μM protein in 25 mM Tris-HCl pH 7.4 and 100 mM NaCl was incubated at 37 °C with 200 r.p.m. orbital shaking. Each reaction was supplemented with 0.01% NaN$_3$ to prevent bacterial growth. At 8 h (for early aggregates/oligomers) or 20–24 h incubation (fibrils or small fibrils), 10 μl of aggregation mixture was collected and stored at 4 °C.

The aβ$_{42}$ protein solutions were prepared from monomeric solutions of human aβ$_{42}$ (Anaspec, San Jose, CA) and prepared by dissolving the lyophilized peptides in SSPE buffer (150 mM NaCl, 10 mM Na$_2$H$_2$PO$_4$ × H$_2$O, and 10 mM Na$_2$EDTA, adjusted to pH 12 using NaOH), followed by sonication over ice for 30 min, and subsequently flash freezing into 5 μL aliquots. Before each of the incubations, aliquots of each peptide were diluted into SSPE buffer (pH adjusted to 7.4 using HCl) to 500 nM (monomer concentration) and placed under conditions for aggregation (37 °C, agitation). Samples were collected after 5 h incubation (for early aggregates/oligomers) or at 20–24 h incubation (for fibrils and small fibrils). The samples were stored on ice and imaged immediately. Data are representative of two aggregation experiments conducted on separate days.

**Cell culture.** All culture reagents were purchased from Invitrogen (UK) unless otherwise specified. Human SH-SY5Y epithelial cells (ATCC CRL-2266) were cultured in Dulbecco's Modified Eagle's Medium (DMEM), F-12 Ham with 25 mM HEPES and NaHCO$_3$ (1:1). Human HEK 293 epithelial cells (ATCC CRL-1573) were cultured in low-glucose DMEM. The base cell culture medium was supplemented with 10% (v/v) fetal bovine serum, 50 units ml$^{-1}$ penicillin, 50 mg ml$^{-1}$ streptomycin and 2 mM L-glutamine.

Cells were cultured in tissue culture flasks (Greiner Bio-One, USA) at 37 °C in 5% CO$_2$ and 95% relative humidity. Cultures were routinely split (1:3) at ∼80% confluency and released for sub-cultivation using 0.25% (v/v) trypsin-EDTA.

**Preparation of cells for imaging.** Sterilized glass coverslips (No. 1 25 mm round, VWR international, Germany) were placed in 6-well tissue culture plastic plates (Greiner Bio-one, USA) and seeded with SH-SY5Y cells or HEK 293 cells at a density of 2 × 10$^6$ cells per well. Cells were cultured overnight and then either imaged live or fixed using 4% paraformaldehyde (Thermo scientific, USA) and 0.2% glutaraldehyde (G6257) in phosphate buffered saline buffer (PBS, Invitrogen) for 30 min before imaging. The coverslip was placed in a holder (Invitrogen), transferred to the microscope stage and imaged at room temperature in an imaging buffer of DMEM without fetal bovine serum, supplemented with 0.05 μM of NR. For SH-SY5Y imaging, at least 20 different cells were imaged on 6 different days.

The cholesterol content of HEK 293 cells was altered as previously described[31]. Briefly, the cholesterol content of cells was increased or decreased by

supplementing the serum-free culture media with methyl-β-cyclodextrin − cholesterol complex at 0.5 mg ml$^{-1}$ for 3.0 h or with methyl-β-cyclodextrin at 2.0 mM for 30 min at 37 °C, respectively. The cholesterol manipulation was confirmed using a cholesterol assay kit (ab133116, abcam, UK). Cells were rinsed twice with warmed PBS and imaged immediately using imaging buffer. Data are representative of at least two experiments conducted on separate days.

**Instrumentation.** Fluorescence imaging was performed using a home-built, bespoke inverted optical microscope (Olympus, IX73) coupled to an electron multiplied charged coupled device (EMCCD) camera (Evolve II 512, Photometrics, Tuscon, AZ). The microscope was configured to operate in objective-type total internal reflection fluorescence (TIRF) mode (for LUVs and protein aggregates) or using highly inclined and laminated optical sheet (HILO) microscopy[46] (for the cell membrane experiments), with the circularly polarized beam of either: 1.) a 100 mW 532 nm continuous wavelength diode-pumped solid-state laser (LASOS Lasertechnik GmbH, Germany) or 2.) a 100 mW 405 nm CW diode laser (Cobolt, MLD 0405-06-01-0100-100) used as light sources. The lasers were directed off a dichroic mirror Di02-R532-25x36 for 532 nm illumination (Semrock, USA) or Di02-R405-25x36 (Semrock, USA) for 405 nm illumination through a high numerical aperture, oil-immersion objective lens (Plan Apochromat 60 × NA 1.49, Olympus APON 60XOTIRF, Japan) to the sample coverslip. Total internal reflection was achieved by focusing the laser at the back focal plane of the objective, off axis, such that the emergent beam at the sample interface was near-collimated and incident at an angle greater than the critical angle $θ_c \sim 67°$ for a glass/water interface for TIRF imaging and slightly less than $θ_c$ for HILO. This generated a ∼50 μm diameter excitation footprint with power densities in the range ∼0.5 kW cm$^{-2}$ at the coverslip. The emitted fluorescence was collected through the same objective and further filtered using a longpass filter BLP01-532R-25 (Semrock, USA) and a bandpass filter FF01-650/200-25 (Semrock, USA) for 532 nm illumination or a longpass filter (FF02-409/LP-25, Semrock, USA) for 405 nm illumination before being expanded by a 2.5 × relay lens (Olympus PE 2.5 × 125). Finally, a physical aperture (VA100/M, Thorlabs) and a transmission diffraction grating (300 Grooves/mm 8.6° Blaze Angle - GT13-03, Thorlabs) were mounted on the camera port path before the detector. The camera-to-grating distance was optimized using TetraSpeck beads (0.1 μm, T7297, Invitrogen) such that undiffracted (zeroth) and first order diffraction was visible on the same image frame in accordance with equation 1. Where the fluorescence image was finally projected onto the EMCCD running in frame transfer mode with at 20 Hz, with an electron multiplication gain of 250, operating at −70 °C with a pixel size of 16 μm and automated using the open source microscopy platform Micromanager.

Transmission Diffraction grating equations were used to estimate the spectral dispersion from the zeroth order at a given distance[47].

$$d\sinθ_m = mλ \quad (1)$$

where $d$ is the grating line spacing, $θ_m$ is the diffraction angle at order $m$, $m$ is the order of diffraction and $λ$ is the wavelength.

$$d\sin(-2γ) = mλ \quad (2)$$

where $γ$ is the blaze angle or the angle between the surface structure and the surface parallel.

**sPAINT Guide.** We detail below the three steps required to generate md-SR sPAINT images: Step 1 calibrates for instrument variation, Step 2 explains how to acquire optimal data, and Step 3 describes how to reconstruct and render the final sPAINT images. For each of these steps, accompanying sample code and data are provided in Supplementary Data 1-5 (see below) and ref. 48.

**Step 1. Spectral calibration, Aberration Correction and Absolute Wavelength Calibration.** The location of the spectral domain on the EMCCD chip is strongly dependent on the orientation and position of the diffraction grating, and so to ensure that the wavelengths determined were correct and therefore comparable, the instrument was calibrated before imaging each day, and following re-alignment.

To calibrate the instrument, TetraSpeck beads were imaged by exciting at 405 nm and the emission collected from 480 to 760 nm (Supplementary Fig. 1a, Supplementary Data 1) (100 frames were collected with a frame rate of 35 ms). The positions of the beads within the spatial domain of the image were determined using the PeakFit plugin (an imageJ/Fiji plugin of the GDSC Single Molecule Light Microscopy (SMLM) package (http://www.sussex.ac.uk/gdsc/intranet/microscopy/imagej/gdsc_plugins) for imageJ (ref. 49) using a typical 'signal strength' threshold of ∼30. The fit results values are then saved as a text file in the directory containing the image for use the subsequent spectral analysis below.

The spectral component of the image (Supplementary Fig. 1b) was analysed using a custom written macros. For non-linear least-squares data fitting the analysis was performed in either; (A) Igor Pro (Wavemetrics) using the Levenberg-Marquardt algorithm to search for the minimum value of chi-square or (B) In the case of the included ImageJ Plugin using the inbuilt fit (Fit.doFit) function (see Supplementary Data 2). With the filter set allowing collection of wavelengths between 480 and 760 nm, there are three clear peaks in the spectral region of the image for each bead (Supplementary Fig. 1a). The position of these three peaks

allows both for the aberration correction (due to the relationship between these and the spatial positions of the beads), and for the pixel-to-wavelength ratio to be determined. The approximate distance between the bead position and each one of the three peaks ($Z_0Z_1$ distance) is be estimated, by the user, and input as the parameters for the imageJ plugin (examples of the three distances are indicated by the green, orange and red arrows in Supplementary Fig. 1a). For each localization (with co-ordinates $x$, $y$, frame number), the spectrum for each of the three peaks was plotted by using the estimated distance and extracting an intensity profile 15 pixels either side of it (in the $y$-direction), and averaging this over a width of 3 pixels in the $x$-direction (to account for the spectra being wider than one pixel). Each intensity profile was fit to a Gaussian distribution to determine the center positions of the peaks (Supplementary Fig. 2). Fits that gave negative amplitudes, centers outside of the range, widths < 1.5 pixels or > 20 pixels were discarded.

The output data were then clustered using R (https://cran.r-project.org). Emission wavelengths at peak maximum ($\lambda$) were set as 512.7, 581.5 and 676.5 nm (Supplementary fig. 3a). $Z_0Z_1$ distances were fitted to a multiple linear regression model:

$$y = \beta_0 + \beta_1 \lambda + \beta_2 x + \beta_3 y + \varepsilon \qquad (3)$$

where $y$ is the $Z_0Z_1$distance (in pixels), $\beta_0$ is the intercept and $\beta_1$ to $\beta_3$ are the regression parameters associated to $\lambda$, $x$ and $y$ dimensions, respectively. These parameters are required for the spectral analysis of datasets.

The corrected wavelength can finally be calculated from $Z_0Z_1$distance according to:

$$\lambda = \frac{1}{\beta_1}(y - \beta_0 - \beta_2 x - \beta_3 y) \qquad (4)$$

R code is available as a script in Supplementary Data 3.

**Step 2. Image acquisition, processing and spectral analysis.** Images were acquired at a frame rate typically < 50 ms and 1,000 frames were acquired for LUVs and protein aggregates, 1,000–3,000 frames for the mammalian cell membrane. (Example 100 nm LUV data are available in ref. 48).

Analogous to the spectral calibration (see above), the co-ordinates corresponding to the localizations within the spatial part of the images were determined using the peak fit plugin for imageJ. For the aggregates and LUVs, the localizations corresponding to each individual unit were clustered using the DBSCAN (ref. 50) algorithm in R (fpc package https://CRAN.R-project.org/package=fpc) using a size of the epsilon neighbourhood (eps) of 0.5 pixels, and a minimum points threshold in the eps of 10. Only clustered localizations were analysed further. Typically, < 20 NR localizations were obtained in a background image ($10.9 \times 10.9\,\mu m$ area) and were not considered as clusters using the same DBSCAN parameters used for data in Figs 2 and 3.

The sPAINT analysis macro (see Supporting code, Supplementary Data 4) with beta parameters (determined in step 1–1 $x$ and $y$) was run using clustered localizations file to retrieve the spectral component of the image.

**Step 3. sPAINT Rendering.** sPAINT rendering was performed using a homemade written plugin for ImageJ (Supplementary Data 5). Each localization was rendered using a numerical erf-based approximation of a discretized 2D Gaussian (integral normalized to unity) with widths in $x$ and $y$-directions depending on localization precision[51,52]. The spectral information at each pixel was calculated as the average of wavelengths observed at this particular pixel weighted by their respective localization intensities and pixels were finally falsely jet-coloured as an overlay over the localization density image.

For analysis of experimental data, an ImageJ plugin and example sPAINT raw data of LUVs are provided[48].

**Instrument stability.** The achievable spatial localization precision, $\sigma$, of the instrument was measured by imaging diffraction-limited TetraSpeck beads (0.1 µm, T7279, Invitrogen) using a 532 nm excitation laser. 2,000 frames at 10 ms per frame were collected at a range of excitation powers and localized using the PeakFit plugin for ImageJ from the GDSC SMLM package. Beads localized in at least half the total number of recorded frames were analysed. TetraSpeck 'orange' peak centres, at ∼580 nm, were fitted and used to determine the spectral localization precision $\sigma$. The same analysis was also use to evaluate the localization precision for the LUV data (Supplementary Fig. 6).

Precision analysis was conducted using MATLAB software and custom written code. The localizations and corresponding spectral centres were grouped using basic cluster analysis, where localizations within a $4 \times 4$ pixel square were considered to originate from the same fluorescent bead/LUV. A histogram of the $x$ and $y$ positions of the localizations within each bead cluster was used to determine the mean position of the bead and the variation in localization position. Each histogram was fit with a 1D Gaussian function in $x$ and $y$ and the wider of the two used to calculate the precision of the localizations, which is equal to the precision of the fitted Gaussian. The mean intensity of each bead/LUV grouping, in photons, was calculated as the mean signal outputted by PeakFit for localizations in each group. PeakFit calculates the intensity of each localization as the integral of the fitted Gaussian (in ADU) divided by the total camera gain (ADU/photon), given by

GainTotal = GainCamera × GainEM × QE. The mean precision values were binned every 200 photons below values less than 2,000 photons and binned every 500 photons above and fit to a single exponential decay using the software OriginPro (version 9, Originlab).

**Data availability.** sPAINT example data is available from the Apollo - University of Cambridge Repository with the identifier doi:10.17863/CAM.6463 (ref. 48). All other data that support the findings of this study are available from the corresponding author upon reasonable request.

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

## Acknowledgements

This work was supported by grants from the MRC (D.K., MR/K015850/1), EPSRC, Royal Society University Research Fellowship (S.F.L., UF120277) and the Augustus Newman Foundation (M.H.H.). We acknowledge the Cambridge Advanced Imaging Centre for the TEM images and Christ's College (M.H.H.). We thank the members of the Klenerman and Lee research groups for their input and discussion, in particular we would like to thank Franziska Kundel, Jason Sang and Dr Janet Kumita for their helpful contributions.

## Author contributions

M.N.B., J.G. and M.H.H. generated and processed data. L.T.T., D.C.W. and J.V.F. supplied test samples. A.R.C. performed the stability analysis. J.-E.L. and A.P. performed analysis. M.N.B., J.G. M.H.H., C.M.D., R.T.R., D.K. and S.F.L. designed experiments. M.N.B., J.G., M.H.H. and S.F.L. wrote the manuscript, and all the authors were involved in the editing and approved the final draft of the manuscript.

## Additional information

**Competing financial interests:** The authors declare no competing financial interests.

