## [Peer Review File · Nature Communications]

Transferred manuscripts:

Reviewers' Comments:

Reviewer #2 (Remarks to the Author):

In their revised manuscript, Bongiovanni et al. have added additional references, made minor textual modifications and performed an additional control experiment using a non-spectrally sensitive membrane binding probe. As noted in the previous review (and confirmed in the authors' response), the novelty of this manuscript lies mainly in the combination of three previously described techniques (Nile red to measure hydrophobicity, PAINT microscopy, and spectral imaging), each of which have been demonstrated separately before, but not in combination. In light of this, aside from a few minor technical questions, my previous comments were largely intended to increase the impact of the manuscript to a general audience by showing new aspects of biology that can be revealed by the technique. I disagree with the author's response to my previous comments #4 and #6.

Re: comment #4 - I don't understand the authors claim that multi-label imaging cannot currently be done serially due to spectral overlap of the probes. Sequential multi-label imaging with identical fluorophores tagged to different epitopes has already been demonstrated with minimal crosstalk by simply washing out the first probe prior to adding a new label (Jungmann et al. Nature Methods, 2014, Kiuchi et al. Nature Methods 2015). Even if the initial probes bound irreversibly, crosstalk could be minimized by photobleaching the vast majority of the initial probe prior to adding a new label. Indeed, serial labeling with multiple probes is one of the key advantages of PAINT microscopy as opposed to other localization microscopy techniques.

Re: comment #6 - I disagree that showing multiple labels in the same sample or mapping hydrophobicity relative to biological structures, "would detract from the key findings and principle of this manuscript". Indeed a large number of localization microscopy papers (including those using PAINT listed above) demonstrate multilabel imaging to reveal the relations of multiple structures within cells. I disagree that showing more biological context for the data would detract from the manuscript.

Although these above concerns are largely unchanged from the initial submission, I do feel that the data in the manuscript support the feasibility of the technique. With the textual changes, the authors have addressed the majority of my previous technical concerns and I believe that the manuscript is likely suitable for publication in Nature Communications. I have only a few minor concerns that should be addressed prior to publication. These are listed below:

1)The PAINT acronym should be described at the first instance of its use. I don't believe that a general reader will know what the letters in PAINT stand for.

2)Please add the size of the scale bar to the caption for supplementary video 1.

3)In figure 4d, the authors claim to visualize "super-resolved hotspots on the plasma membrane". I'm unclear what is actually being super-resolved here. The features in the dataset and movie appear to be on the order of 2-3 microns. I see no hotspots that are below the diffraction limit. Given the amount of spatial and temporal smoothing required to obtain a high enough localization density this is not particularly surprising. The authors should either clarify what structures below the diffraction limit

are being observed (thus super-resolved), or remove this statement from the text.

4) Moreover, the legends states that figure 4d probes the temporal membrane dynamics in a fixed cell. Can the authors please clarify what exactly is the biological relevance of these dynamics? Are they simply stochastic fluctuations in dye binding or do they represent stochastic lipid dynamics (since these may not be completely fixed by formaldehyde). Is there any kind of control that could discriminate between these two? In absence of a biological control, a simulated dataset could be constructed to verify that hot spots do not arise simply from stochastic dye binding.

Reviewer #3 (Remarks to the Author):

The revised paper by Dr Lee and co-workers addressed several of my comments, it is generally improved, acknowledges the state of the art and the combination of spectral sensitive detection with PAINT using environment sensitive dyes is novel and of substantial interest. I still have a slight disagreement about some replies to my previous comments, but in general I support publication in Nature Communications.

1. Previous comment 1: spatial and spectral resolution. In their response, the authors state that background does not influence the resolution, since photon counts above the background are used. This is not correct. Due to the shot-noise of the background, it deteriorates the localization precision and thus the spatial and spectral resolution (see e.g. R. E. Thompson, D. R. Larson, and W. W. Webb, "Precise nanometer localization analysis for individual fluorescent probes," *Biophys J*, vol. 82, p. 2775, 2002; K. I. Mortensen, L. S. Churchman, J. A. Spudich, and H. Flyvbjerg, "Optimized localization analysis for single-molecule tracking and super-resolution microscopy.," *Nat Methods*, vol. 7, no. 5, pp. 377-381, May 2010.). Thus under imaging conditions with low background (bright beads, low laser powers, no NR in solution) the localization precision will be much better than for NR with the same photon numbers above background, but higher background. As the theoretical localization precision can be calculated from photons and background, the authors could plot those curves for NR, using measured photon and background values. As the spectral localization precision directly derives from the spatial localization precision, also this value could be calculated for realistic NR parameters. In my view, this point is important, since later in the manuscript conclusions are drawn from the width of the peak position histograms, which depends on the spectral precision.

2. Previous comment 3: analysis of LUV data. I still think that the size measurements of LUVs with sPAINT as described here are not accurate. With a residence time of > 100 ms (Figure 2f,g), many NR molecules will be fluorescent during the frame time of 50 ms. Assuming a diffusion coefficient of NR in lipid membranes of $1 \mu\text{m}^2/\text{s}$, the molecules move on average 400 nm during the frame time, therefore, they explore the whole LUV. Thus localizing them results in an averaged position located near the center of the LUV. This is reflected in the images, where the LUVs have an apparent size comparable to the scale bar, and thus below their real size. The analysis using 95% of the total intensity as a binary cutoff is somewhat arbitrary and could overestimate the size. As the measurement of the size of the LUV with sPAINT does not contribute to the core message of the paper, it could be removed.

Reviewer comments shown in blue, our responses in black, changes to the manuscript have been underlined

Reviewer #2 (Remarks to the Author):

In their revised manuscript, Bongiovanni et al. have added additional references, made minor textual modifications and performed an additional control experiment using a non-spectrally sensitive membrane binding probe. As noted in the previous review (and confirmed in the authors' response), the novelty of this manuscript lies mainly in the combination of three previously described techniques (Nile red to measure hydrophobicity, PAINt microscopy, and spectral imaging), each of which have been demonstrated separately before, but not in combination. In light of this, aside from a few minor technical questions, my previous comments were largely intended to increase the impact of the manuscript to a general audience by showing new aspects of biology that can be revealed by the technique. I disagree with the author's response to my previous comments #4 and #6.

Once again, we wish to thank the reviewer for their positive comments. Our detailed responses to them are highlighted below:

Re: comment #4 - I don't understand the authors claim that multi-label imaging cannot currently be done serially due to spectral overlap of the probes. Sequential multi-label imaging with identical fluorophores tagged to different epitopes has already been demonstrated with minimal crosstalk by simply washing out the first probe prior to adding a new label (Jungmann et al. Nature Methods, 2014, Kiuchi et al. Nature Methods 2015). Even if the initial probes bound irreversibly, crosstalk could be minimized by photobleaching the vast majority of the initial probe prior to adding a new label. Indeed, serial labeling with multiple probes is one of the key advantages of PAINt microscopy as opposed to other localization microscopy techniques.

The reviewer is correct; it is possible to do multi-label PAINt imaging with dyes that have spectral overlap by simply following each imaging round with a washing step before introducing the next label. This is advantageous to the technique developed here, and would allow other properties to be observed simultaneously on the same sample indeed we have already published doing this previously, albeit in a non-spectrally resolved way (doi: 10.1073/pnas.1114444108). We have therefore added a sentence to the main text highlighting this major advantage of the PAINt technique (page 13):

“It should also be possible to monitor multiple properties in the same sample either by using probes that have no spectral overlap, pre imaging/bleaching of non-PAINt probes, or by performing sequential imaging rounds interspersed with washing steps to remove each PAINt label.”

Re: comment #6 - I disagree that showing multiple labels in the same sample or mapping hydrophobicity relative to biological structures, "would detract from the key findings and principle of this manuscript". Indeed a large number of localization microscopy papers (including those using PAINt listed above) demonstrate multilabel imaging to reveal the relations of multiple structures within cells. I disagree that showing more biological context for the data would detract from the manuscript.

As highlighted in the reviewer's previous comment, it should be possible to perform multiple rounds of imaging on the same sample by washing out the PAINt dye after each imaging step, as has been demonstrated previously with PAINt. We agree that this could be useful, and we be the topic of a future study. However, the aim of this manuscript is to show that the

method can be used to investigate biological properties, such as hydrophobicity, at the nanoscale. The manuscript convincingly shows this, and we believe that showing multi-label imaging does not add anything substantial to the manuscript, but could detract the reader from the principal aim. Indeed, spectrally-resolved super-resolution for multi-labeling has already been demonstrated and we have already included these references in the existing manuscript (reference 10-12).

Although these above concerns are largely unchanged from the initial submission, I do feel that the data in the manuscript support the feasibility of the technique. With the textual changes, the authors have addressed the majority of my previous technical concerns and I believe that the manuscript is likely suitable for publication in Nature Communications. I have only a few minor concerns that should be addressed prior to publication. These are listed below:

1) The PAINT acronym should be described at the first instance of its use. I don't believe that a general reader will know what the letters in PAINT stand for.

We thank the reviewer for pointing this out, and have now added the acronym, shown below:

“Here, we report the development of the first md-SR imaging technique termed spectrally-resolved PAINT (points accumulation for imaging in nanoscale topography¹³) or sPAINT”

2) Please add the size of the scale bar to the caption for supplementary video 1.

The scale bar in Supplementary Video 1 is 1 μm , and we have now added this to the figure caption.

3) In figure 4d, the authors claim to visualize "super-resolved hotspots on the plasma membrane". I'm unclear what is actually being super-resolved here. The features in the dataset and movie appear to be on the order of 2-3 microns. I see no hotspots that are below the diffraction limit. Given the amount of spatial and temporal smoothing required to obtain a high enough localization density this is not particularly surprising. The authors should either clarify what structures below the diffraction limit are being observed (thus super-resolved), or remove this statement from the text.

We thank the reviewer drawing our attention to a mistake in the Figure 4d legend; the scale bar is not 10 μm but 1 μm (the legend has been corrected). The representation of the data using temporal and spatial smoothing tends to reduce the amplitudes of spectral variations, and therefore does not allow us to precisely define the shape and size of domains of differing hydrophobicity at this current time. However it is clear that some cross-sections of these domains are below (or roughly equal to) the diffraction limit of our instrument (see some examples below).

Response Figure 1. Counts vs. distance traces from the dashed lines in the sPAINT images (left panels) are presented to the right. Arrows in each case highlight regions in which sub-diffraction hotspots are shown.

4) Moreover, the legends states that figure 4d probes the temporal membrane dynamics in a fixed cell. Can the authors please clarify what exactly is the biological relevance of these dynamics? Are they simply stochastic fluctuations in dye binding or do they represent stochastic lipid dynamics (since these may not be completely fixed by formaldehyde). Is there any kind of control that could discriminate between these two? In absence of a biological control, a simulated dataset could be constructed to verify that hot spots do not arise simply from stochastic dye binding.

It is generally thought that lipids are poorly fixed by paraformaldehyde + 0.2% glutaraldehyde and membranes are thought to remain 'fluid' after fixation (Kasumi *et al.* Nat Methods. 2010, 7(11):865-6.). However, we do agree with reviewer 2 that the biological relevance of lipid dynamics in fixed cells may be limited, and that their membrane dynamics are likely to differ from their those in live cells.

Nevertheless, much work has focused on studying membrane biophysics of passively

controlled (Giant Unilamellar Vesicles, LUVs) or extracted (Giant Plasma Membrane Vesicles) lipid mixtures – providing valuable information about membrane properties (Baumgart et al PNAS 2007, Sezgin et al Nat Prot 2012, Veatch et al Chem Bio 2008, for example). Our intention in this work was to determine the limits of sPAINT, and to validate whether it was able to visualize membrane heterogeneity arising from compositional fluctuations present in cell membranes. Although we cannot report the precise quantitative characterization of membrane domains at this time, nor provide a sophisticated theoretical model based on our initial observations, we have shown that sPAINT can be used for the spatio-temporal characterization of membrane dynamics, and that it can be used by others to research such systems.

In order to determine whether the hotspots occur due to stochastic dye-binding, we have generated a simulated dataset. This was achieved by comparing our experimental data directly with a complete spatial randomness model (*i.e.* random x and y positions generated according to a spatial Poisson process) and then resampling the spectral values for each (x,y) pair with a random value of λ selected from the list of initially observed spectral values. These new data are now included in Response Video 1. As expected, the sustained hotspots that are represented as dark-blue hydrophobic regions (highlighted in Figure 4d in the manuscript) are no longer observed.

Reviewer #3 (Remarks to the Author):

The revised paper by Dr Lee and co-workers addressed several of my comments, it is generally improved, acknowledges the state of the art and the combination of spectral sensitive detection with PAINT using environment sensitive dyes is novel and of substantial interest. I still have a slight disagreement about some replies to my previous comments, but in general I support publication in Nature Communications.

1. Previous comment 1: spatial and spectral resolution. In their response, the authors state that background does not influence the resolution, since photon counts above the background are used. This is not correct. Due to the shot-noise of the background, it deteriorates the localization precision and thus the spatial and spectral resolution (see e.g. R. E. Thompson, D. R. Larson, and W. W. Webb, "Precise nanometer localization analysis for individual fluorescent probes," *Biophys J*, vol. 82, p. 2775, 2002; K. I. Mortensen, L. S. Churchman, J. A. Spudich, and H. Flyvbjerg, "Optimized localization analysis for single-molecule tracking and super-resolution microscopy.," *Nat Methods*, vol. 7, no. 5, pp. 377-381, May 2010.). Thus under imaging conditions with low background (bright beads, low laser powers, no NR in solution) the localization precision will be much better than for NR with the same photon numbers above background, but higher background. As the theoretical localization precision can be calculated from photons and background, the authors could plot those curves for NR, using measured photon and background values. As the spectral localization precision directly derives from the spatial localization precision, also this value could be calculated for realistic NR parameters. In my view, this point is important, since later in the manuscript conclusions are drawn from the width of the peak position histograms, which depends on the spectral precision.

The reviewer is correct; at low photon counts, the shot-noise from the background is able to influence the precision/resolution of the method. The intention of performing the analysis on TetraSpeck beads was to determine the upper limit for the spatial precision of the instrument (not necessarily working with PAINT dyes). For this, we found that we could obtain an ultimate precision of 6.6 ± 0.2 nm. We wish to draw the reviewer's attention to Supplementary Figure 6, in which analogous analysis was performed on **nile red labeled LUVs** (Supplementary Figure 6), giving rise to an ultimate precision of around 8 nm. This was performed under identical imaging conditions to those in all other sPAINT experiments in the manuscript, and serves to highlight that we are able to obtain a high spatial, and

therefore spectral precision even under conditions in which there is a higher background.

As the reviewer points out, it is possible to theoretically determine the localization precision; however, we (and others in the field) have found that this method to overestimate the actual precision (see below), and therefore believe that the empirically-determined spatial precision is more correct in this instance. We have included a direct comparison of this effect below in Response Figure 2.

Response Figure 2. Comparison of the theoretical and empirically-determined spatial precision as a function of the integrated photon number. The Watt W. Webb equation prediction has been used to theoretically determine the localization precision using values directly obtained from exciting TetraSpeck dyes (yellow curve). However, it overestimates the localization precision (as it does not take into account any engineering limitation of the microscope) compared to the actual measured values by ~2-3 nm throughout typical photon ranges for both fluorescent proteins and organic dyes (500-1500 and 1500-3500 photons respectively), specifically it overestimates by 2.0 nm (12%) at 500 photons, 1.9 nm (21%) at 1500 photons and 1.8 nm (23%) at 3500 photons.

2. Previous comment 3: analysis of LUV data. I still think that the size measurements of LUVs with sPAINT as described here are not accurate. With a residence time of > 100 ms (Figure 2f,g), many NR molecules will be fluorescent during the frame time of 50 ms. Assuming a diffusion coefficient of NR in lipid membranes of $1 \mu\text{m}^2/\text{s}$, the molecules move on average 400 nm during the frame time, therefore, they explore the whole LUV. Thus localizing them results in an averaged position located near the center of the LUV. This is reflected in the images, where the LUVs have an apparent size comparable to the scale bar, and thus below their real size. The analysis using 95% of the total intensity as a binary cutoff is somewhat arbitrary and could overestimate the size. As the measurement of the size of the LUV with sPAINT does not contribute to the core message of the paper, it could be removed.

We agree with the reviewer that using a 95% intensity cut-off threshold may be somewhat arbitrary, and have therefore re-analysed the data by fitting the LUVs in the super-resolution images to 2D Gaussian distributions, and using the measured FWHM as an indication of the LUV diameter. With this analysis, we find that the median LUV diameter of 90 nm, which is

comparable to the dynamic light scattering measurement of the same sample (diameter of 105 nm). We believe this is an improved method for analysing the data and have now replaced SI Fig 7 with an alternative one using this new sizing method suggested by reviewer 3.